# Altered myocardial lipid regulation in junctophilin-2–associated familial cardiomyopathies

Satadru K Lahiri[1,2] , Feng Jin[3], Yue Zhou[4], Ann P Quick[1,2] , Carlos F Kramm[1,2], Meng C Wang[3,4,5], Xander HT Wehrens[1,2,6,7,8,9]

Myocardial lipid metabolism is critical to normal heart function, whereas altered lipid regulation has been linked to cardiac diseases including cardiomyopathies. Genetic variants in the *JPH2* gene can cause hypertrophic cardiomyopathy (HCM) and, in some cases, dilated cardiomyopathy (DCM). In this study, we tested the hypothesis that JPH2 variants identified in patients with HCM and DCM, respectively, cause distinct alterations in myocardial lipid profiles. Echocardiography revealed clinically significant cardiac dysfunction in both knock-in mouse models of cardiomyopathy. Unbiased myocardial lipidomic analysis demonstrated significantly reduced levels of total unsaturated fatty acids, ceramides, and various phospholipids in both mice with HCM and DCM, suggesting a common metabolic alteration in both models. On the contrary, significantly increased di- and triglycerides, and decreased coenzyme were only found in mice with HCM. Moreover, mice with DCM uniquely exhibited elevated levels of cholesterol ester. Further in-depth analysis revealed significantly altered metabolites from all the lipid classes with either similar or opposing trends in *JPH2* mutant mice with HCM or DCM. Together, these studies revealed, for the first time, unique alterations in the cardiac lipid composition—including distinct increases in neutral lipids and decreases in polar membrane lipids—in mice with HCM and DCM were caused by distinct *JPH2* variants. These studies may aid the development of novel biomarkers or therapeutics for these inherited disorders.

## Introduction

Junctophilin-2 (JPH2) is a structural protein that connects the plasma membrane (PM) to intracellular organelles such as the endo/sarcoplasmic reticulum (ER/SR) in striated muscle cells (Nishi et al, 2000). Inherited variants in the *JPH2* gene have been shown to cause hypertrophic cardiomyopathy (HCM), a potentially lethal disorder characterized by left ventricular hypertrophy and an elevated risk of cardiac arrhythmias (Landstrom et al, 2007). Less commonly, inherited *JPH2* variants can also cause dilated cardiomyopathy (DCM), which causes thinning and enlargement of the left ventricle, as well as potentially lethal arrhythmias (Lehnart & Wehrens, 2022).

The functional consequences of inherited *JPH2* variants associated with cardiomyopathy remain incompletely understood. Initial studies using an H9C2 rat cardiomyoblast cell line showed that HCM-associated *JPH2* variants induced hypertrophic growth (Landstrom et al, 2007). Our laboratory demonstrated that the A405S missense variant (A399S in mice) identified in patients with HCM causes aberrant organization of transverse tubules, abnormal intracellular Ca2+ handling, and histological signs of structural remodeling (i.e., myocyte hypertrophy, fibrosis) (Quick et al, 2017). Although some *JPH2* variants associated with DCM have been identified in patients, such as the truncation variant E641* (Jones et al, 2019), little is known about the potential molecular mechanisms that cause cardiomyopathy development. Interestingly, the position of the E641 residue is the same in both mouse and human JPH2 proteins.

In excitable cells, PM/ER contact domains are known to play important roles in Ca2+ signaling and lipid homeostasis, among other things (Li et al, 2021). JPH2 tethers the transverse tubules to the PM and was shown to recruit functional L-type Ca2+ channels to lipid rafts in adult cardiomyocytes (Poulet et al, 2021). The "membrane occupancy and recognition nexus" (MORN) repeats in the JPH2 protein are believed to interact with lipids, in particular phospholipids in the PM. Palmitoylation of JPH2 enables binding to lipid raft domains in the PM (Jiang et al, 2019). Moreover, Prisco et al (2023 *Preprint*) recently showed that fatty acid oxidation is impaired in human induced pluripotent stem cell–derived cardiomyocytes in which *JPH2* was ablated. A recent lipidomic study of surgical myectomy specimens from an HCM patient with symptomatic left

[1]Cardiovascular Research Institute, Baylor College of Medicine, Houston, TX, USA   [2]Department of Integrative Physiology, Baylor College of Medicine, Houston, TX, USA   [3]Molecular and Human Genetics, Baylor College of Medicine, Houston, TX, USA   [4]Huffington Center on Aging, Baylor College of Medicine, Houston, TX, USA   [5]Howard Hughes Medical Institute, Baylor College of Medicine, Houston, TX, USA   [6]Department of Medicine, Baylor College of Medicine, Houston, TX, USA   [7]Department of Neuroscience, Baylor College of Medicine, Houston, TX, USA   [8]Department of Pediatrics, Baylor College of Medicine, Houston, TX, USA   [9]Center for Space Medicine, Baylor College of Medicine, Houston, TX, USA

Correspondence: wehrens@bcm.edu

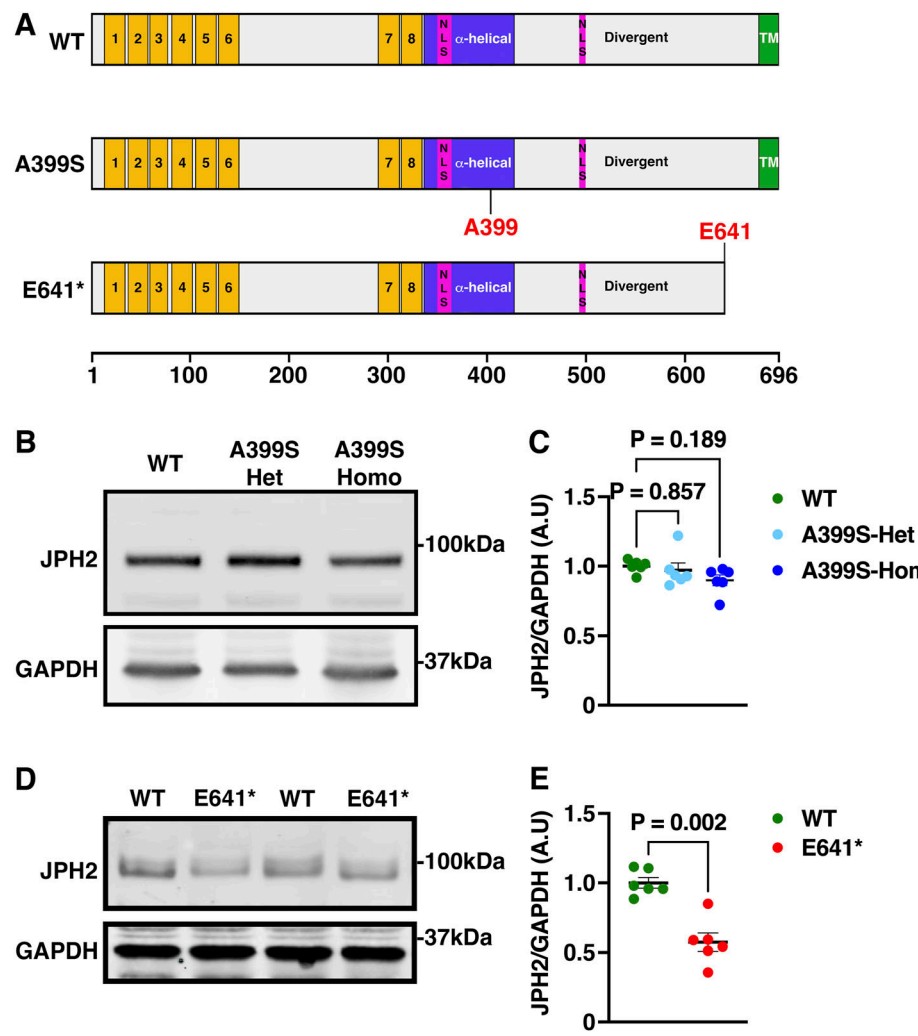

**Figure 1. JPH2 protein levels in JPH2 mutant mouse hearts.**
**(A)** Schematic cartoon of the mouse JPH2 protein showing WT, A399S missense variant, and the truncating E641* variant. Yellow domains (1–8) mark the "membrane occupancy and recognition nexus" domains; pink domains mark nuclear localization sequences; and green domains mark the transmembrane (TM) domain. **(B, C)** Representative Western blot images (B) and quantification of JPH2 protein levels normalized to GAPDH levels in hearts from A399S-Het and A399S-Homo mutant mice (C). N = 6 in each group. One-way ANOVA was used to determine potential statistical differences between the three groups. **(D)** Representative Western blot images of JPH2 and GAPDH in hearts from E641* mutant mice. NS indicates non-specific protein bands in WB. **(E)** Quantification of JPH2 protein levels normalized to GAPDH levels in hearts from E641* mutant mice. N = 6 in each group. The Mann–Whitney test was used to calculate the statistical significance between the two groups. Average values are represented as the mean ± SEM.
Source data are available for this figure.

ventricular outflow tract obstruction revealed accumulation of free fatty acids, whereas the concentrations of acylcarnitines and free carnitine were markedly reduced (Ranjbarvaziri et al, 2021). Given the scarcity of knowledge in this area, our main objective was to study how *JPH2* variants associated with different types of cardiomyopathies (i.e., HCM and DCM) impact lipid metabolism in the heart.

In this study, we conducted unbiased myocardial lipidomic profiling of hearts from novel knock-in mice with *JPH2* variants A399S and E641* found in patients with HCM and DCM, respectively. Clear differences were identified in the cardiac lipid signatures comparing WT mice with those carrying the A399S versus E641* variant in *JPH2*, respectively. Certain lipid groups showed similar changes in both mutants compared with WT mice, whereas other lipids changed in opposite directions depending on the specific *JPH2* variant. For example, there was accumulation of di- and triglycerides in A399S mutant mice with HCM, whereas these lipid groups were depleted in E641* mutant mice with DCM. To the best of our knowledge, this is the first study to report unbiased myocardial lipid profiles from different familial cardiomyopathies caused by

variants in the same gene. This study expands our understanding of the potential role of JPH2 in cardiac lipid metabolism. Moreover, an in-depth analysis of these differential lipid signatures may be critical to identifying potential biomarkers and therapeutic targets for HCM and DCM.

# Results

## Characterization of cardiac dysfunction in JPH2 mutant mice with cardiomyopathy

To uncover the mechanism underlying distinct cardiomyopathy phenotypes caused by *JPH2* monogenic variants, we developed two new mouse alleles harboring different *JPH2* variants, that is, missense variant A399S (A405S in human JPH2, associated with HCM) and truncation variant E641* (associated with DCM). The A399S variant is located in the alpha-helical region of JPH2, which is thought to span the dyad between the PM and ER/SR (Quick et al, 2017) (Fig 1A). The E641* variant causes a stop codon near the

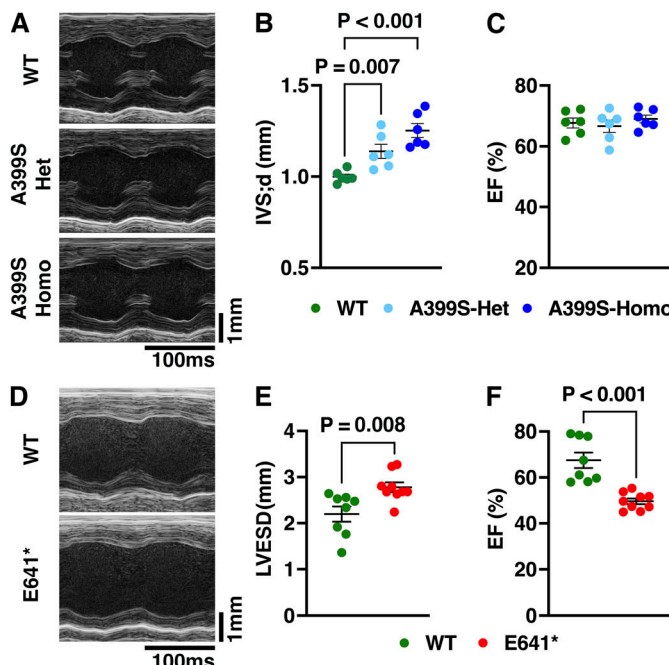

**Figure 2. JPH2 variants in knock-in mice cause different types of cardiomyopathy.**
**(A)** Representative M-mode echocardiography images from A399S-homozygous (Homo), A399S-heterozygous (Het), and WT littermate controls. **(B, C)** Quantification of the intraventricular septum (IVS) thickness in diastole (B) and ejection fraction (EF) (C) in A399S mutant mice compared with WT controls. N = 6 in each group. One-way ANOVA was used to calculate the statistical significance between the three groups. **(D)** Representative M-mode echocardiography images from E641* Het and WT controls. **(E, F)** Quantification of left ventricular end-systolic diameter (E) and ejection fraction (EF) (F) in E641* mutants compared with WT controls. N = 8 in WT and 9 in the E641* group. The Mann–Whitney test was used to calculate the statistical significance between the two groups. All values are represented as the mean ± SEM.

C-terminal end of JPH2 in the divergent domain (Jones et al, 2019). Both alleles were generated using CRISPR/Cas9 genome editing (see the Materials and Methods section). Both A399S-heterozygous (Het) and A399S-homozygous (Homo) mice were viable and followed the Mendelian ratio during breeding. We validated the mutations in these models by Sanger sequencing (Fig S1A and B). Interestingly, we never obtained any E641* Homo mice when we crossed E641* Het mice, suggesting this allele is lethal in the homozygous state.

To determine the effects of the variants on JPH2 protein expression levels, Western blotting was performed using left ventricular tissue from both mouse lines. JPH2 protein levels were unaltered in A399S-Het (0.97 ± 0.05, $P$ = 0.857) and A399S-Homo (0.90 ± 0.04, $P$ = 0.189) mice compared with WT littermate controls (1.00 ± 0.02 arbitrary units) (Fig 1B and C). On the contrary, JPH2 protein levels were significantly reduced in the hearts of E641* Het mice (0.57 ± 0.06) compared with WT controls (1.00 ± 0.04, $P$ = 0.002) (Fig 1D and E). These latter findings suggest that E641* Homo mice may not be viable because of a complete loss of JPH2.

Next, we assessed cardiac function in the mutant mice to determine whether it mimics the clinical phenotype observed in

human patients carrying the corresponding *JPH2* variants (Quick et al, 2017; Jones et al, 2019). Echocardiography of 8-mo-old A399S-Het and A399S-Homo mice revealed HCM characteristics including a significantly increased intraventricular septal thickness (1.14 ± 0.04 mm in Het, $P$ = 0.007 versus WT; 1.25 ± 0.04 mm in Homo, $P$ < 0.001 versus WT; compared with 1.00 ± 0.01 mm in WT). The ejection fraction (EF) was unaltered in the A399S-Het (66.7% ± 2.04%, $P$ = 0.690) and A399S-Homo (69.1% ± 1.3%, $P$ = 0.541) mice compared with WT littermates (67.8 ± 1.6) (Fig 2A–C; Table S1). Echocardiography of 12-mo-old E641* Het mice revealed DCM characteristics with a significantly increased end-systolic left ventricular internal diameter (2.78 ± 0.11 mm in E641* versus 2.19 ± 0.16 mm in WT, $P$ = 0.008) and significantly reduced EF (49.6% ± 1.2% in E641* versus 67.5% ± 3.4% in WT, $P$ < 0.001) (Fig 2D–F, Table S1). Thus, both JPH2 mutant mouse alleles are accurate models of the clinical phenotype observed in human variant carriers.

## Cardiac lipidomic signatures in JPH2 mutant mice with cardiomyopathy

To uncover potential changes in the cardiac lipidomic landscape because of the *JPH2* variants, we collected the tissue (30 mg) from the same location—left ventricle from A399S-Homo, A399S-Het, E641* Het, and WT controls (n = 4 for each group). Unbiased lipidomic profiling of these samples was performed using a thermo UPLC coupled with a high-resolution Orbitrap Lumos as outlined in Fig 3A–D. The lipids were identified and quantified based on accurate MS1 and data-dependent MS/MS with Thermo LipidSearch software. Next, we assessed the overall change in lipid groups in these models. Lipids detected in our lipidomic analysis were classified into five groups and 28 classes (Fig 3E).

A principal component analysis conducted on total lipids from the heart samples revealed clustering among the four genotypes of mice (Fig 4A). The ellipsoid around each group indicates the 95% confidence interval. A clear separation between the A399S and E641* mutant mice was observed, whereas the WT group intersected both mutants. There was no significant change in the relative distribution of lipid groups normalized to the total amount of lipids for each group (Fig 4B). The pie charts show that phospholipids were most abundant (91.8–92.1%), followed by fatty acyls (6.53–6.92%), sphingolipids (1.06–1.18%), glycerolipids (0.15–0.20%), and sterol lipids (0.0026–0.0034%).

To assess whether there were changes in the amounts of lipid classes within the five major groups, we plotted the normalized changes for each of the three mutant groups (A399S-Het, A399S-Homo, and E641* Het) compared with the WT groups (Fig 4C). Within the fatty acyl group, total unsaturated fatty acid (FA) was significantly decreased in all groups (−45.1% ± 1.4% in A399S-Het, −50.1% ± 6.6% in A399S-Homo, and −52.8% ± 5.9% in E641* compared with WT controls, $P$ = 0.008, 0.004, and 0.002, respectively) (Fig 4C), whereas acylcarnitine and anandamide were not significantly changed.

Among the glycerolipids, triglycerides (TG) were significantly increased in A399S mice (+145.3% ± 47.5% in A399S-Het and +172.5% ± 18.0% in A399S-Homo compared with WT controls, $P$ = 0.016 and 0.005, respectively, Fig 4C). The diglycerides (DG) were also significantly upregulated in A399S-Homo knock-in mice (+74.0% ± 9.8% in A399S-Homo compared with WT controls, $P$ = 0.014). There was a trend

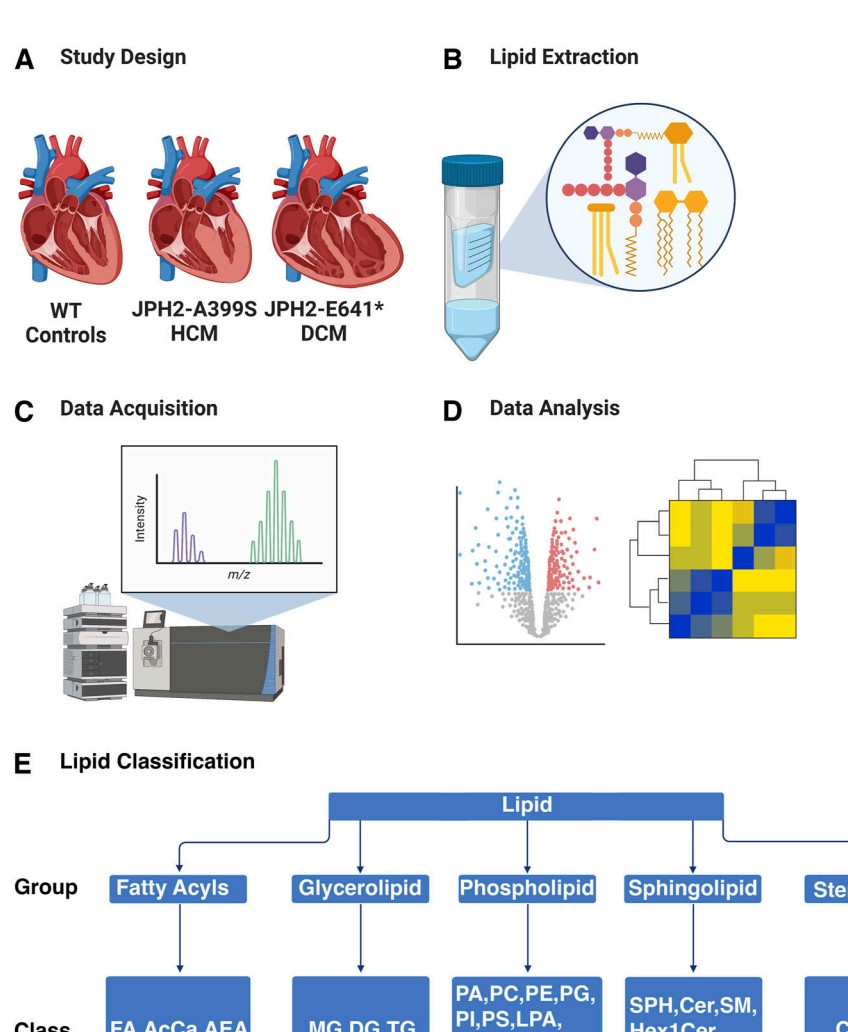

**Figure 3. Schematic flowchart of the lipidomic screening and analysis.**
**(A)** Hearts were obtained from WT controls, JPH2-A399S-Het, JPH2-A399S-Homo mutant mice, and JPH2-E641* Het mutant mice. **(B)** Lipids were extracted from cardiac tissue samples of these groups of mice using methanol, methyl tert-butyl ether, and water as described in this study. **(C)** Lipidomic analysis was performed using a Vanquish UPLC and an Orbitrap Lumos mass spectrometer (Thermo Fisher Scientific Inc.). **(D)** Data analysis was performed using LipidSearch and BioPAN. **(E)** Lipids were classified into five categories (fatty acyls, glycerolipids, phospholipids, sphingolipids, and sterol esters) and 28 classes (abbreviations are shown below these five categories). Our analysis identified a total of 1,656 lipid metabolite subclasses (this level of detail is not shown here).

toward increased DG in A399S-Het knock-in mice as well (+55.8% ± 24.2% in A399S-Het compared with WT controls, *P* = 0.067). On the contrary, the monoglycerides (MG) were decreased (−40.8% ± 3.4% in A399S-Het compared with WT controls, *P* = 0.034). MG was also decreased in A399S-Homo mice although it was not significant (−26.3% ± 9.2% in A399S-Homo mice compared with WT controls, *P* = 0.22). These results suggest an increased DG and TG biosynthesis from MG in A399S mice. Interestingly, there was a trend toward reduced MG, DG, and TG levels in E641* mice, although this was non-significant.

Among the phospholipids, we observed significantly decreased levels of lysophosphatidic acid (LPA), lysophosphatidylcholines (LPC), and lysophosphatidylethanolamine (LPE) in all the groups. LPA was reduced by −71.3% ± 2.6% in A399S-Het, −71.1% ± 1.5% in A399S-Homo, and −70.6% ± 2.8% in E641* mice compared with WT controls, *P* < 0.001 for all groups. LPC was reduced by −44.4% ± 3.6% in A399S-Het, −43.1% ± 1.8% in A399S-Homo, and −45.9% ± 1% in E641* mice compared with WT controls, *P* < 0.001 for all groups.

Moreover, LPE was reduced by −37.9% ± 2.7% in A399S-Het, −38.7% ± 4.7% in A399S-Homo, and −43.8% ± 2.3% in E641* mice compared with WT controls, *P* < 0.001 for all groups. These results suggest the significant impact of JPH2 deficiency on lysophospholipid metabolism. In addition, we also observed that phosphatidic acid (PA) was only significantly down-regulated in A399S-Het mice (−32% ± 9.3% in A399S-Het compared with WT controls, *P* = 0.006) (Fig 4C). Similarly, phosphatidylcholines (PC) were only significantly decreased only in A399S mice (−7.5% ± 2% in A399S-Het and −8.8% ± 1.4% in A399S-Homo compared with WT controls, *P* = 0.010 and 0.003, respectively). Together with the reduction of LPA and LPC in A399S mice, these results revealed that the A399S variant leads to increased catabolism of PA and PC. On the contrary, the total levels of other phospholipid classes (PE, PG, PI, PS, lysophosphatidylglycerol, lysophosphatidylinositol, CL, and mono-lysocardiolipin) were unchanged in all groups compared with WT controls.

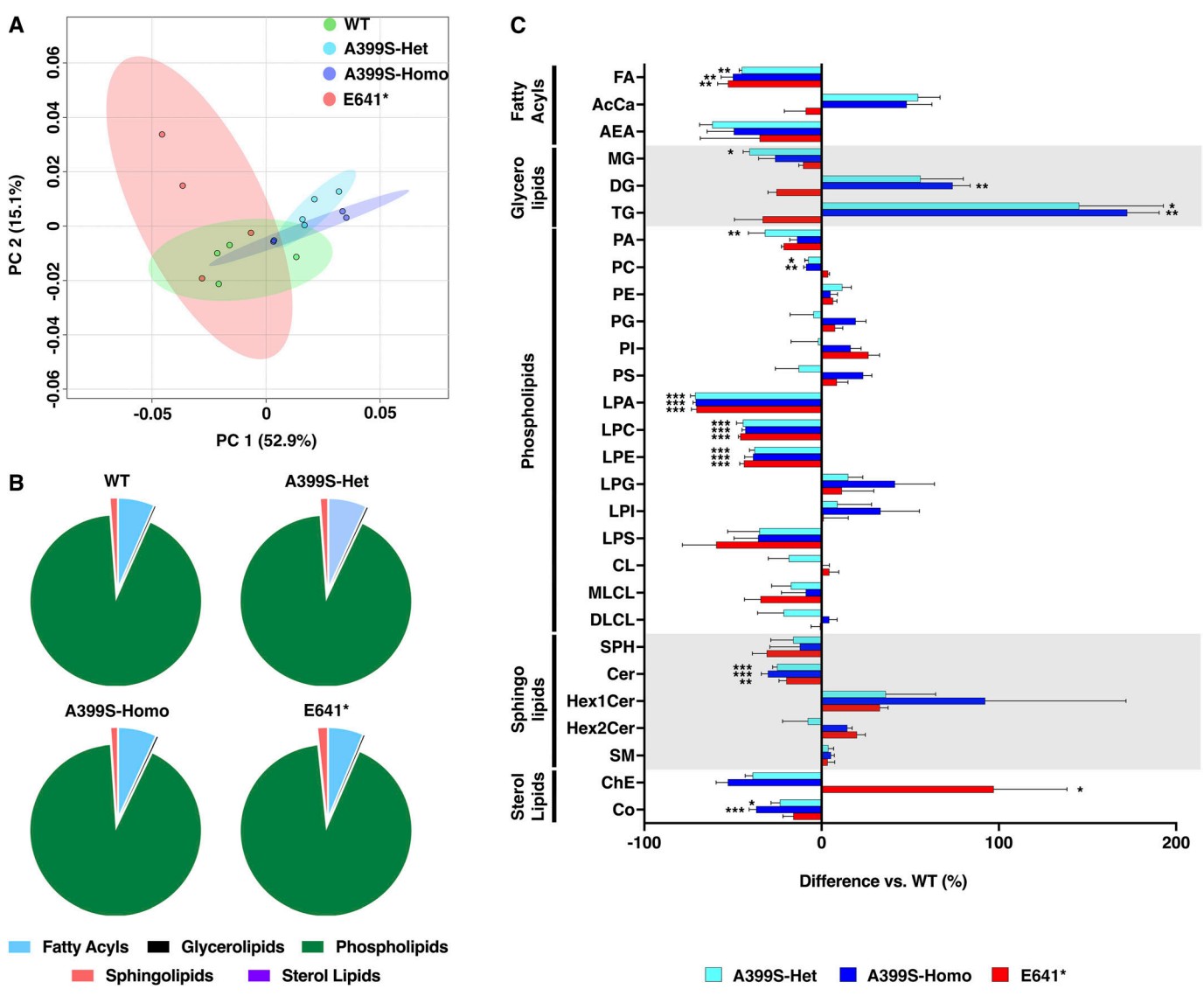

**Figure 4. Altered lipid profiles in the hearts of JPH2 mutant mice.**
**(A)** Principal component (PC) analysis showing the variance in the lipidomic data. PC1 accounted for 52.9%, and PC2 accounted for 15.1% of the variance. **(B)** Pie charts showing the distribution of major lipid groups (i.e., fatty acyls, glycerolipids, phospholipids, sphingolipids, and sterol lipids) in WT, A399S-Het, A399S-Homo, and E641* Het mice. **(C)** Bar graphs demonstrating changes in the levels of different classes of lipids in A399S-Het, A399S-Homo, and E641* Het mouse hearts compared with WT littermate controls. Individual lipid classes were normalized to the level in WT mice. All values are represented as the mean ± SEM. One-way ANOVA was performed, and the adjusted $P$-value is presented for each group compared with WT controls. *$P < 0.05$, **$P < 0.01$, and ***$P < 0.001$.

Among the sphingolipids, ceramides (Cer) were significantly reduced in all the groups (−25.1% ± 2.5% in A399S-Het, −30.3% ± 3.7% in A399S-Homo, and −19.9% ± 4.2% in E641* mice compared with WT controls, $P < 0.001$, $P < 0.001$, and $P = 0.004$, respectively) (Fig 4C). Among the sterol lipids, cholesterol ester (ChE) was significantly increased by +97.1% ± 41.4% in E641* mice compared with WT controls, $P = 0.031$ (Fig 4C). In addition, co-enzyme (Co) was significantly decreased only in the A399S model (−23.5% ± 5% in A399S-

Het and −36.9% ± 4% in A399S-Homo compared with WT controls, $P = 0.019$ and $P < 0.001$, respectively).

## Changes in JPH2 mutant mice with cardiomyopathy

Further analysis led to the identification of a total of 1,656 lipid metabolites within the 28 classes of lipids (Fig 3E). The fold change in each lipid metabolite was calculated for each of the three mutant

mouse lines compared with WT mice, and lipid metabolite cluster analysis was performed to assess the hierarchy of the overall changes.

Next, we evaluated all altered lipid metabolites again by ranking them based on the fold change value, after setting a cutoff fold change value of 2 (i.e., a fold change of either >2 or <−2) and a statistically significant P-value of <0.05 versus WT. Separate volcano plots were created for each genotype (Fig 5A–C). A399S-Het mice had 202 up-regulated and 83 down-regulated lipid metabolites in the heart (Fig 5A), whereas A399S-Homo mice had 256 up-regulated and 112 down-regulated lipid metabolites (Fig 5B). Moreover, E641* mice had 39 up-regulated lipid metabolites and 98 down-regulated lipid metabolites (Fig 5C).

Two heatmaps were generated of the top 25 most significantly altered lipid metabolites, one comparing A399S-Het and A399S-Homo versus WT (sorted by Homo P-value; Fig 6A) and the other comparing E641* Het versus WT (sorted by P-value; Fig 6B). TG(16:0e_18:2_18:2), PC(36:3e), and PS(41:7) were the three most significantly decreased metabolites, whereas TG(15:0_16:0_18:2), DG(28:2e), and PC(20:5_20:5) were the three most significantly increased metabolites in the A399S HCM model (Fig 6B). In mice with the E641* variant, the three most significantly decreased metabolites were PA(44:6), CL(84:8), and PS(41:7), whereas the three most significantly increased metabolites were PC(20:5_20:5), PE(48:3), and TG(15:0_18:1_15:0). Finally, we identified 10 common metabolites (PS(41:7), CL(84:8), PS(43:7), CL(80:8), PE(20:4e), PC(20:5_20:5), LPA(22:6), PE (19:0_22:6), LPE (22:4), and PC (39:5)) in the top 25 lists for both models, all of which showed a similar directional trend in both models.

Next, we analyzed all metabolites within each lipid class and selected the top five metabolites with either a >2 fold change or <−2 fold change compared with the WT controls. We then combined the top five metabolites from each group, removing duplicates, and calculated row z-scores to produce heatmaps that evaluated metabolite changes across different groups. Lipid classes with no significant lipid metabolite changes were excluded. The findings presented from lipid metabolites in the fatty acyl group (Fig S2A and B), glycerolipids (Fig S3A and B), phospholipids (Fig S4A–N), sphingolipids (Fig S5A–E), and sterol lipids (Fig S6) show similar changes between the A399S-Het and A399S-Homo mice, whereas the E641* showed different lipid metabolite changes, indicating a difference between these two models in higher resolution. In addition, we observed some similarly altered metabolites in both the HCM and DCM models. Overall, the lipid metabolite profiles in the hearts of A399S and E641* mutant mice are extensively altered, suggesting potential disease-specific metabolic remodeling.

### Lipidomic pathway analysis

Finally, to gain better insight into the mechanisms underlying the altered lipid profiles in these JPH2 mutant disease models, the overall myocardial lipidomic data were analyzed using MetaboAnalyst software (Xia et al, 2009) (Fig 7A, C, and E). Both models showed a significant difference in lipid enrichment, with glycosphingolipids being most enriched in the A399S model of HCM, and glycerophosphates being primarily enriched in the E641* model of DCM. Although there were differences in the enrichment signatures comparing the different models, some similarities were also observed. For example, ceramides were somewhat more enriched in the A399S model compared with the E641* mutant, and glycerophosphoglycerol was specifically increased in the E641* DCM model.

We further assessed the lipid network reactions in BioPAN (Gaud et al, 2021), as biological alterations usually yield complex changes in multiple lipid metabolites. Fig 7B, D, and F represents BioPAN lipid networks from A399S-Het, A399S-Homo, and E641* hearts, respectively. The nodes denote the lipid classes, and each arrow between the two nodes demonstrates the reaction direction. The green arrow indicates a positive Z-score, whereas the purple arrow depicts a negative Z-score. The most active reactions in the A399S-Het model were ceramide (Cer) to sphingomyelin (SM) (Z-score = 2.97) followed by phosphatidic acid (PA) to diglycerol (DG) (Z-score = 2.14), which indicates a significantly increased sphingomyelin biosynthesis and diglycerol accumulation in these hearts. On the contrary, biosynthesis of phosphatidylcholine (PC) and cardiolipin (CL) was suppressed in this model. Interestingly, the A399S-Homo model showed similarly increased SM biosynthesis (Cer to SM, Z-score = 2.69) and DG accumulation (PC to DG, Z-score = 2.7, and PA to DG, Z-score = 1.85) as observed in the A399S-Het model, further strengthening the role of JPH2 in lipid regulation. The A399S-Homo mice also exhibited active PA-to-PS reaction (Z-score = 2.59), PA-to-PI reaction (Z-score = 2.52), and PC-to-PS reaction (Z-score = 2.25). Similar to the Het model, we found suppressed PG-to-CL reaction (Z-score = −3.54). Overall, both the A399S-Het and A399S-Homo models of HCM showed increased biosynthesis of sphingomyelin, diglycerol, and phosphatidylserine and a reduced biosynthesis of phospholipids such as phosphatidylcholine (PC) and phosphatidylethanolamine (PE).

In contrast, the most active reactions in the E641* model of DCM were PA to PI (Z-score = 3.65) and DG to PC (Z-score = 2.35). Similar to the HCM model, we also found a significant increase in SM biosynthesis in E641* mice (Cer to SM, Z-score = 2.04). LPC-to-LPA (Z-score = −3.65) and PC-to-DG (Z-score = −2.21) reactions were significantly suppressed in the DCM model. The most exciting finding was the difference in DG and PC reactions comparing the HCM and DCM models. We found active DG metabolism in the HCM model, whereas DG was actively converted to other phospholipids such as PE, PC, and PA in the DCM model.

## Discussion

In this study, we performed unbiased lipidomic profiling of the hearts from two mouse models of distinct familial cardiomyopathies (HCM and DCM) caused by variants in the same JPH2 gene. Major findings were as follows: (i) JPH2-A399S knock-in mice develop an HCM phenotype similar to the human carriers of this variant; (ii) JPH2-E641* knock-in mice develop a DCM phenotype with systolic dysfunction similar to the human patients carrying this JPH2 truncation variant; (iii) there were significantly reduced levels of total unsaturated fatty acids, ceramides, and various phospholipids in both mouse models compared with WT controls; (iv) significantly increased di- and triglycerides and decreased coenzyme were only found in JPH2-A399S mice with HCM; and (v)

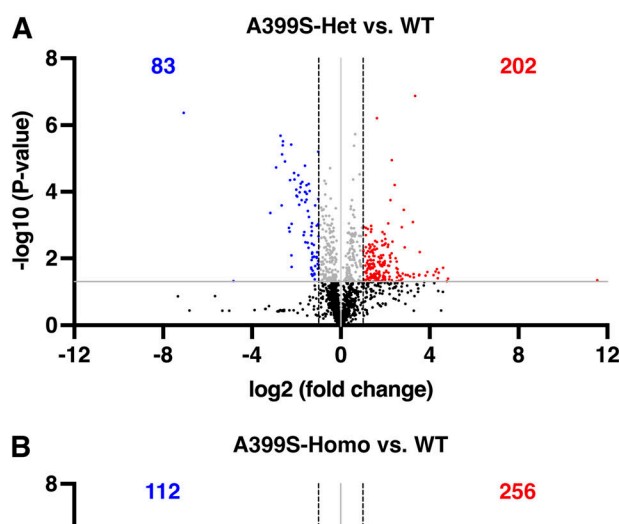

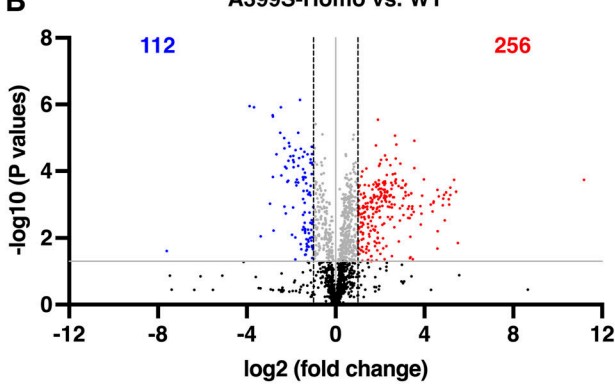

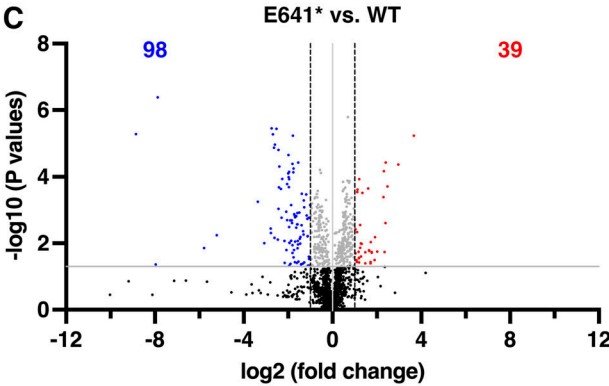

- ● Significantly upregulated > 2-Fold (P<0.05)
- ● Significantly downregulated > 2-Fold (P<0.05)
- ● Significant changes < 2-Fold
- ● Nonsignificant changes

**Figure 5. Fold changes in lipid metabolites in different JPH2 mutant mouse models.**
**(A, B, C)** Volcano plots representing overall fold changes in lipid metabolites in A399S-Het (A), A399S-Homo (B), and E641* mice (C) compared with WT littermate controls. The horizontal gray line indicates a *P*-value of 0.05. Dots above the gray line indicate statistically significant fold changes in lipid metabolites. The vertical black dashed lines are placed at the cutoff fold changes of 2 (log (fold change) of 1) and −2 (log (fold change) of −1). Red dots indicate significantly increased lipid metabolites (>2 fold change), and blue dots indicate significantly reduced lipid metabolites (<−2 fold change). The number of significantly changed metabolites above the cutoff fold change is listed on each volcano plot in red (increased) and blue (decreased).

JPH2-E641*mice with DCM uniquely exhibited elevated levels of cholesterol ester. Based on these results, JPH2 pathogenic mutations are associated with increased neutral lipids (DG and TG in A399S mice, and ChE in E641* mice) and decreased polar membrane lipids (phospholipids and sphingolipids). To our best knowledge, this is the first study showing distinct cardiac lipidomic signatures in two types of familial cardiomyopathies caused by variants in the same gene. These studies may aid the development of novel biomarkers or therapeutics for these inherited disorders.

Lipids play a crucial role in cardiomyocyte membrane structure and are essential for maintaining normal cardiac function (Park et al, 2007). In addition, fatty acids are the primary factor in cardiac metabolism and energy production (Lopaschuk et al, 2021). On the contrary, alterations in the lipid content of the heart have been associated with cardiac disease (Schulze et al, 2016). Moreover, increased plasma levels of cholesterol and lipoproteins can lead to coronary artery disease. Recent developments in mass spectrometry–based unbiased lipidomic screening techniques allow for high-throughput screening of different lipid molecules in plasma and tissues with higher resolutions. A large-scale lipidomic study in rats revealed cardiac lipids to be categorized into five groups, including fatty acyls, glycerolipids, phospholipids, sphingolipids, and sterol lipids (Pradas et al, 2018). There was high abundance of the lipid classes of free fatty acids (FA), diglycerides (DG), triglycerides (TG), phosphatidylcholine (PC), phosphatidyl-ethanolamine (PE), cardiolipin (CL), and ceramides (Cer).

Myocardial lipidomics allows for the identification and quantification of specific lipid classes and metabolites in the heart that are significantly altered in cardiac disease states. For example, increased levels of certain lipid classes, such as DG, TG, and Cer, have been implicated in myocardial inflammation, oxidative stress, and apoptosis, which are hallmarks of ischemic heart diseases (Tomczyk & Dolinsky, 2020; Shu et al, 2022). Conversely, decreased levels of certain lipid classes, such as PC, PE, and CL, have been associated with impaired mitochondrial function and cardiac metabolism, which are key deficits in heart failure (Shen et al, 2015; Wittenbecher et al, 2021). Although plasma lipids have been evaluated as possible biomarkers for various cardiac diseases, relatively little remains known about changes in the lipid profile in cardiac diseases such as inherited cardiomyopathies, ischemic heart disease, and diabetic cardiomyopathy (Ranjbarvaziri et al, 2021; Li et al, 2022; Gaggini et al, 2023).

Our study focused on two types of inherited cardiomyopathy, HCM and DCM, caused by inherited variants in the JPH2 gene (Quick et al, 2017; Jones et al, 2019). JPH2 is a structural protein that connects the PM to intracellular organelles such as the ER/SR and likely also the mitochondria (Garbino et al, 2009; Beavers et al, 2014; Prisco et al, 2023 *Preprint*). Inherited JPH2 variants associated with HCM cause aberrant transverse tubule organization and deficits in intracellular Ca²⁺ handling (Quick et al, 2017). Interestingly, similar cellular defects were observed in a non-genetic model of pressure overload–induced heart failure, which could be rescued by the overexpression of JPH2 (Reynolds et al, 2016). More recently, we also identified several *JPH2* variants in patients with DCM, including the truncation variant E641*, but little remains known about the pathogenesis of this condition (Jones et al, 2019). Given that a few recent studies suggested that JPH2 interacts with phospholipids in

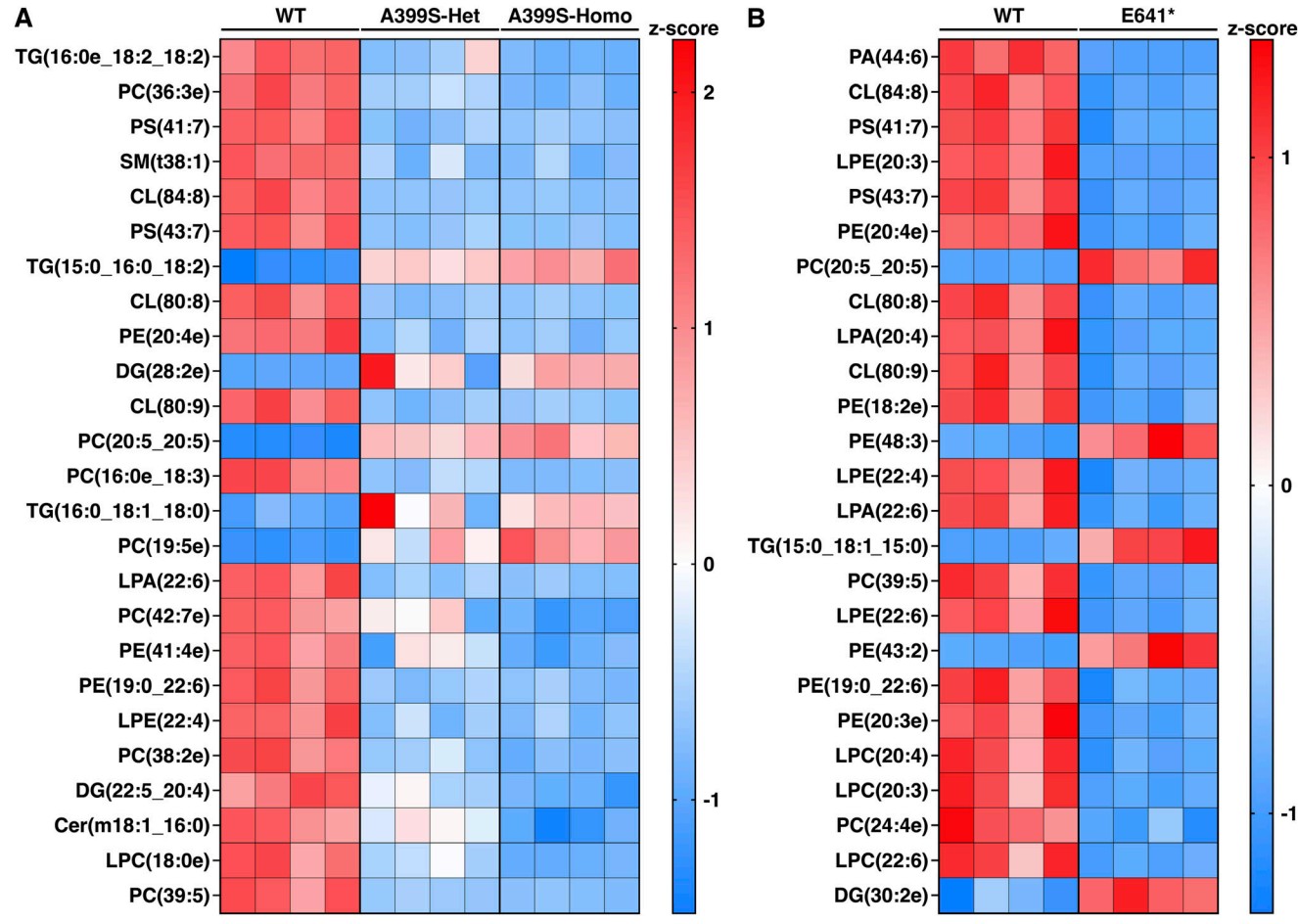

**Figure 6. Alterations in lipid distribution in JPH2 mutant mouse models.**
**(A)** Heatmap showing the top 25 most significantly changed lipid metabolites in A399S-Homo mice compared with Het and WT littermate controls. **(B)** Heatmap showing the top 25 most significantly changed lipid metabolites in E641* Het mice compared with WT littermate controls. Both heatmaps are plotted using row z-scores.

the PM and that JPH2 ablation in human induced pluripotent stem cell–derived cardiomyocytes impaired fatty acid oxidation, we set out to perform a comprehensive lipidomic analysis of two knock-in mouse models of JPH2-induced cardiomyopathies.

We found a significant reduction in FA in both the JPH2-linked HCM and DCM models (Fig 4C). On the contrary, our data did not reveal significant changes in acylcarnitine and anandamide. Prior studies have established a metabolic switch from FA oxidation to glycolysis in failing hearts because of a lack of cardiac FA (Lopaschuk et al, 2021). Furthermore, alterations in FA metabolism impact calcium handling and cardiac remodeling during the progression of heart failure (Willis et al, 2015). Our high-resolution lipidomic profiling identified significantly reduced levels of docosatetraenoic acid (FA [22:4]), docosapentaenoic acid (FA [22:5]), and docosahexaenoic acid (FA [22:6] or DHA) in both cardiomyopathy models. Arachidonic acid (FA [20:4]) was significantly reduced in the HCM model but was unchanged in the DCM model. DHA is an omega-3 long unsaturated fatty acid that along with eicosapentaenoic acid was shown to reduce cardiovascular mortality and improve cardiac function in patients with myocardial infarction or coronary heart disease (Khan et al, 2021). Moreover, higher plasma

concentrations of DHA are associated with a lower risk of developing heart failure (Zheng et al, 2022). Consistently, several studies in patients and animal models have established that DHA has a cardioprotective role. For example, a pressure overload–induced model of cardiac hypertrophy demonstrated that DHA decreases mitochondrial membrane viscosity, improves calcium uptake, and ameliorates disease progression (Dabkowski et al, 2013). The cellular signaling role of DHA was identified in a study with a mouse model of myocardial ischemia. Future studies may reveal whether normalization of FA levels could slow the development of JPH2-associated cardiomyopathy.

In the glycerolipid group, we found significantly increased levels of DG and TG in the JPH2-A399S model of HCM. Interestingly, DG was reduced in the JPH2-E641* model of DCM, whereas TG was unchanged. The BioPAN pathway analysis revealed active DG synthesis and accumulation in the HCM model but not in the DCM model. Prior studies have associated elevated DG and TG plasma with an increased heart failure risk (Quehenberger & Dennis, 2011; Toth et al, 2019). As a secondary messenger, DG actively regulates cell signaling pathways by activating PKC, PKD, RAC GTPase-activating protein, and RAS guanyl nucleotide-releasing proteins

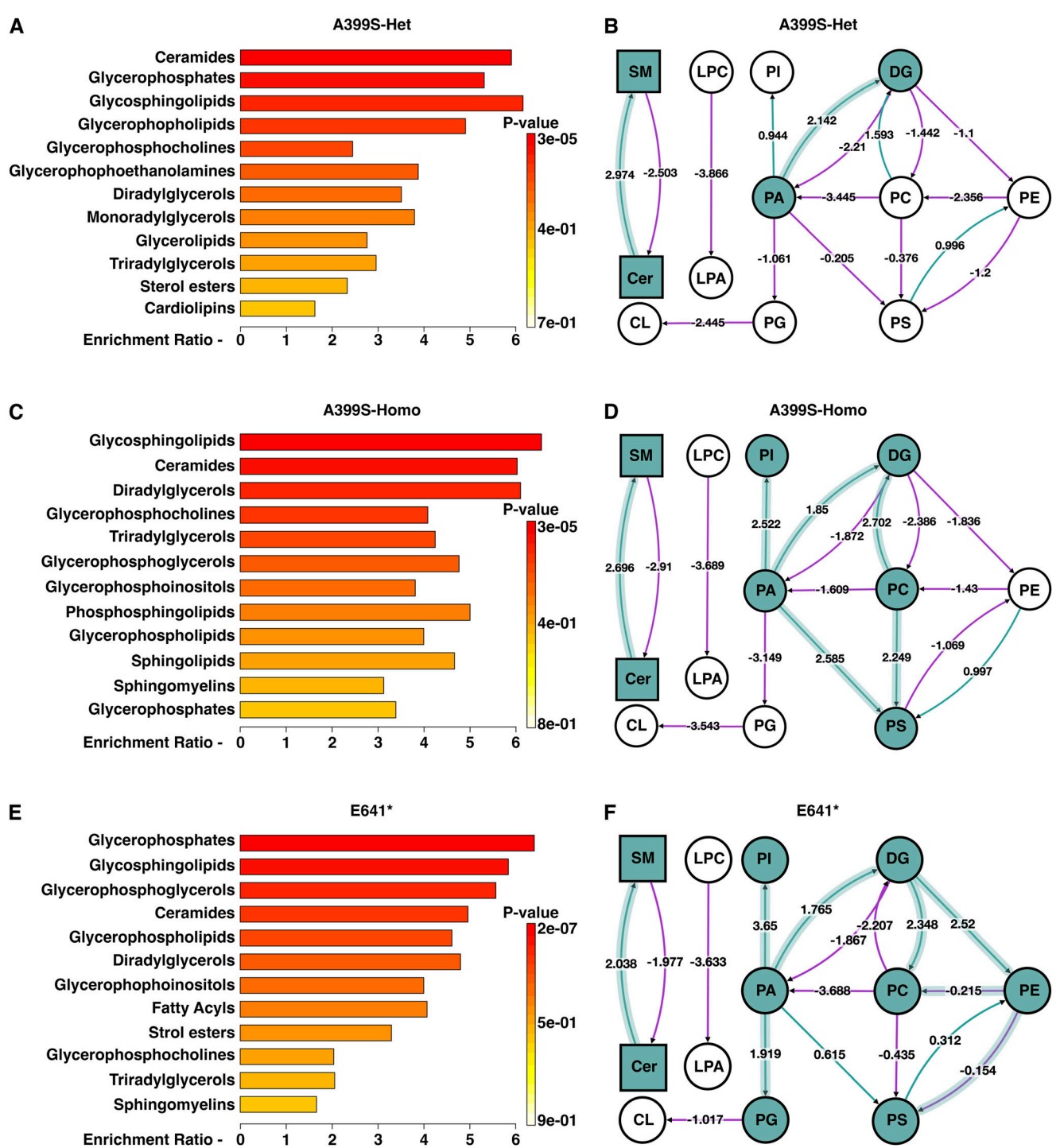

**Figure 7. Lipid metabolite enrichment in JPH2 mutant models.**
**(A, C, E)** Overview of enriched lipid metabolites in A399S-Het (A), A399S-Homo (C), and E641* mice (E) prepared using MetaboAnalyst. **(B, D, F)** Lipid class reaction pathway analysis for A399S-Het (B), A399S-Homo (D), and E641* mice (F) based on comprehensive lipidomic profiling in BioPAN. Green arrows are associated with positive z-scores, and purple arrows are associated with negative z-scores. The direction of the arrows indicates the direction of the reaction.

(Kolczynska et al, 2020). Several studies established that over-activation of PKC promotes calcium mishandling, contractile dysfunction, and cardiac hypertrophy (Singh et al, 2017). Moreover, DG-mediated activation of PKC can trigger increased intracellular calcium levels followed by CaMK hyperactivation leading to cardiac hypertrophy (Wu et al, 2006). In mouse models, DG inhibitor "diacylglycerol kinase" overexpression prevented cardiac hypertrophic remodeling (Arimoto et al, 2006). Therefore, increased DG

levels in the hearts of JPH2-A399S mice suggest that DG-mediated prohypertrophic signaling contributes to the HCM disease phenotype (Quick et al, 2017). Interestingly, DG (22:5/20:4) and DG (18:2/20:4) were significantly reduced, whereas most other DG metabolites significantly increased in the HCM mouse model. Elevated levels of TGs have been observed in patients with diabetes and obesity (Sharma et al, 2004). Moreover, in animal models of obesity and diabetes, TG accumulation in cardiomyocytes triggered lipotoxicity and contractile dysfunction (Young et al, 2002; Unger, 2003). Although JPH2 dysfunction has not been linked to TG synthesis, one GWAS in females with early menarche and metabolic syndrome identified an SNP near the *JPH2* gene locus, which potentially regulates its expression levels (Lee et al, 2019). Finally, a prior lipidomic study of rats with diabetes and DCM identified a reduction in most myocardial PC metabolites, like our observations in the E641* model of DCM (Dong et al, 2017). Follow-up studies are needed to elucidate the role of JPH2 in glycerolipid metabolism in the heart.

In terms of the phospholipid group, we found significantly reduced levels of various phospholipids in both JPH2-associated cardiomyopathy models. Specifically, LPA, LPC, and LPE were significantly reduced in both models, whereas PA and PC were significantly decreased only in the HCM model (Fig 4C). Although the mechanism of interaction of JPH2 binding to phospholipids is somewhat unclear, studies have shown that JPH2 might bind to PS and PI by means of its MORN domains (Bennett et al, 2013). In addition, S-palmitoylation of cysteine residues within the MORN domains has been shown to be essential for JPH2 binding to lipid rafts (Jiang et al, 2019). Another study demonstrated that cholesterol is critical for JPH2 and T-tubule assembly (Poulet et al, 2021), further suggesting that the phospholipid content in the cardiomyocyte membrane could be vital for JPH2 tethering T-tubules and maintaining junctional membrane complex integrity. The abundance of plasma lysophospholipids in heart failure is controversial (Law et al, 2019). Several studies have shown decreased LPC levels in cardiovascular diseases (Lee et al, 2013; Stegemann et al, 2014), whereas increased LPC levels were found within LDL particles (Zakiev et al, 2016).

In the present study, we found reduced levels of various cardiac lysophospholipids in two mouse models of cardiomyopathy. We identified CL (84:8), CL (80:8), and CL (80:9) as the three most significantly reduced lipid metabolites in both cardiomyopathy models, although the overall cardiolipin (CL) contents were unchanged. Moreover, CL's primary fatty acid moiety (18:2) was also significantly reduced in the JPH2-E641* model of DCM.

It is well established that cardiolipin is a critical component of the mitochondrial membrane and that cardiolipin deficiency is associated with impaired mitochondrial function and altered cardiac metabolism leading to cardiac disease (Dudek et al, 2019). Therefore, our result suggests that high-resolution lipidomic screening could potentially find alterations in critical lipid metabolites when the total lipid classes remain unchanged. Consistent with our findings, lipidomic studies of phospholipids in left ventricular samples from adult and pediatric patients with DCM revealed that cardiolipin is inversely associated with disease progression (Sparagna et al, 2007; Chatfield et al, 2014).

Interestingly, CL (18:2) was the primary metabolite reduced in these patients' hearts, like our finding in the JPH2-E641* DCM mutant model.

Our BioPAN pathway analysis revealed significantly enhanced SM biosynthesis in both cardiomyopathy models even though the total amount was not changed. Moreover, we found significantly reduced levels of ceramides (Cer) in both JPH2 mutant models, although several metabolites within the Hex1-Cer and SM classes were actually increased. We observed a significantly increased Cer-to-SM conversion in both mouse models. Ceramides are the precursor for most sphingolipids. Ceramide synthase 5 (CerS5), which mediates the synthesis of Cer (16:0) and Cer (14:0), promotes autophagy and cardiac hypertrophy because of lipotoxicity (Russo et al, 2012). In contrast, we observed decreased levels of Cer (16:0) and Cer (18:0) in our models. Conversely, several studies have established that ceramides promote different cardiac diseases by triggering mitochondrial dysfunction and apoptosis (Choi et al, 2021). Based on our findings, we conclude that the deficiency of ceramides is due to increased conversion of ceramides to SM, although total SM levels were unchanged. The levels of several SM metabolites were significantly increased in both models, although most fold changes were less than our cutoff score (fold change = 2). Future in-depth metabolic flux studies might reveal critical lipid signaling pathways and metabolic switches associated with the disease phenotypes in our mouse models of HCM and DCM.

Overall, we present a comprehensive analysis of the cardiac lipidome in two different cardiomyopathy disease models caused by pathogenic variants in JPH2. To our best knowledge, this is the first study of the cardiac lipidome in defined genetic models of cardiomyopathies. In addition to uncovering altered lipid classes involved in cardiac pathophysiology, we identified differentially altered lipid metabolites in both models. This comprehensive lipid dataset from these cardiac disease models might enable the identification of potential disease biomarkers and provide a potential tool used for the diagnosis of heart failure in the future.

## Materials and Methods

### Animal studies

All studies were performed according to protocols approved by the Institutional Animal Care and Use Committee of Baylor College of Medicine, conforming to the Guide for the Care and Use of Laboratory Animals published by the U.S. National Institutes of Health (NIH Publication No. 85-23, revised 1996). Both JPH2-A399S and JPH2-E641* knock-in alleles were generated using CRISPR/Cas9 with cytosolic Cas9 protein embryo microinjections. Guide RNAs targeting the respective gene locus were designed by Genetically Engineered Rodent Models Core. Mice were generated on a C57BL/6J background supplied by Jackson Laboratories (stock number: 000664) and backcrossed for at least five generations before using in experiments. After four to six backcrosses, the A399S-heterozygous mice were crossed together to get A399S-homozygous mice. All experiments were performed in male mice

to minimize variability in the relatively small number of animals per group (n = 4) for lipidomic analysis.

### Echocardiography

Anesthesia was induced with isoflurane (1.5–2.0% vol/vol in $O_2$) in an induction chamber. After chest hair was removed using Nair cream, mice were placed on a heated platform (38.5°C) to maintain body temperature between 36.5°C and 37.5°C. Anesthesia was maintained using inhaled isoflurane (1.5–2.0% vol/vol in $O_2$). Echocardiography was performed using a Vevo 2100 system with a 30-MHz frequency probe (VisualSonics) as described (Lahiri et al, 2020). First, the parasternal short-axis images were recorded in the B-mode, followed by the M-mode capture. Next, we turned the probe 90° to obtain a parasternal long-axis view in the B-mode. Mice were tilted on their left side to obtain a better view of the intraventricular septum. The short-axis M-mode images were analyzed using Vevo 2100 software to assess systolic function and left ventricle dimensions. The long-axis B-mode images were used to evaluate the intraventricular septum thickness. The echocardiographer was blinded to the mouse identity to avoid bias during data collection.

### Western blotting

Mouse left ventricular samples were suspended in RIPA buffer containing 1.0% CHAPS, Phos-STOP, complete mini protease inhibitor cocktail (Roche), 20 mM NaF, and 1.0 mM Na3VO4, followed by homogenization with steel beads using a homogenizer (TissueLyser LT—QIAGEN) at 50 s$^{-1}$ for 6 min. We sonicated the homogenized tissue three times for 2 s each and centrifuged at 15,500$g$ for 20 min at 4°C. The lysates (supernatant) were collected, and we measured the protein concentration using NanoDrop (Thermo Fisher Scientific). 60 mg of lysate was loaded into the SDS–agarose gel (10%) after diluting with SDS Laemmli buffer (1610737; Bio-Rad) and 5% beta-mercaptoethanol (M3148; Sigma-Aldrich). After transferring the protein onto 0.45-micron polyvinylidene fluoride membranes, membranes were blocked for 1 h at RT using Prometheus blocking buffer (Genesee Scientific). Membranes were incubated with anti-JPH2 and anti-GAPDH antibodies overnight at 4°C after three washes in TBST and incubation with secondary antibody for 1 h at RT. After washing in TBST post–secondary antibody incubation, membranes were developed using an Odyssey infrared imager (Li-COR Scientific). Bands were quantified using ImageJ software and normalized to respective GAPDH levels. We used a JPH2 antibody (40-5300; Invitrogen) with a C-terminal epitope.

### Lipidomic analysis in mouse heart tissue

The mouse LV tissue was crushed with liquid nitrogen, and 30 mg of mouse tissue was homogenized using a bead homogenizer. SPLASH LIPIDOMIX Mass Spec standards (Avanti Polar Lipids) were spiked into each sample before extraction. The samples were extracted using methanol, methyl tert-butyl ether, and water following the extraction method reported before (Yamada et al, 2013). The extracted samples in MTBE were dried and resuspended in methanol and isopropanol (50:50, vol/vol). The samples were analyzed using a Vanquish UPLC and an Orbitrap Lumos mass spectrometer (Thermo Fisher Scientific). The mobile phase A was 5 mM ammonium formate with 0.1% formic acid in water and acetonitrile (50:50, vol/vol), and mobile phase B consisted of 2-propanol, acetonitrile, and water (88:10:2, vol/vol). A Thermo Accucore Vanquish C18+ reverse phase column (Thermo Fisher Scientific) separated the lipids detected in positive and negative ionization modes. Mass spectra were acquired in the full-scan and data-dependent MS2 mode. The resolution for MS1 was 120 k. The mass scan range was 250–1,200; the maximum injection time was 50 ms; the AGC target was 200,000; and the RF lens was 45%. For MS2 scanning, 20 dependent scans were acquired in each cycle. The MS2 resolution was 30 k; HCD was used to fragment precursor ions with stepped collision energy 25, 30, and 35; and the AGC target is 50,000. High-throughput analysis of lipidomic data was performed using LipidSearch 4.2 software (Thermo Fisher Scientific) (Taguchi & Ishikawa, 2010; Yamada et al, 2013). Lipid quantification using precursor ion area and lipid was identified by matching product ion spectra to an in silico LipidSearch library. Both precursor and product ion mass tolerance were set at 5 ppm. The M-score threshold was set at 2.0. Then, both the positive and negative data were aligned based on the retention time tolerance of 0.1 min and mean value. The final data were filtered based on the preferred ion adduct for each lipid class.

### Lipidomic data processing and analysis

Data were exported with the lipid metabolite format from LipidSearch 4.2 software. The negative and positive values were combined for individual lipid metabolites in all groups. Data were further filtered for the highest lipid adduct ion for each lipid molecule. We confirmed that there were no duplicate values for any of the groups. Next, we normalized the peak area by the total peak area for all lipid molecules. The tissue weight factor was already normalized during the lipid extraction process. We averaged all the groups separately and measured fold changes by dividing the A399S-Het, A399S-Homo, and E641* average by the WT average. P-values were calculated between different groups with the fold change data using either $t$ test between two groups or a one-way ANOVA test for more than two groups. The adjusted P-values were plotted on the graphs. We began by analyzing all metabolites in each lipid class. In the first step, we sorted the list according to the adjusted P-value in the A399S-Homo group. Next, we further narrowed down the metabolites in our results to only include those that had a significant fold change of either >2 or <−2 compared with the WT. Next, we organized our data based on the most significant changes in the E641* group. We then applied the same filter criteria as before to only retain metabolites that had a significant change as previously defined. Lastly, we calculated the row z-scores for each metabolite by dividing the difference between the fold change in an individual sample and the average of all samples by the SD of all samples. The row z-scores were then used to create heatmaps to evaluate changes in metabolites across different lipid classes.

## Statistical analysis

Results are expressed as the mean ± SEM. GraphPad Prism was used for an unpaired *t* test or ANOVA after performing the D'Agostino–Pearson normality test for normal data distribution. A one-way ANOVA test with Tukey's multiple comparison test was used for more than two groups. Adjusted *P*-values were listed, and a *P*-value < 0.05 was considered statistically significant.

## Supplementary Information

## Acknowledgements

This work was funded in part by NIH grants AG062257 (to MC Wang) and HL089898, HL147108, HL153350, and HL160992 (to XHT Wehrens), the Howard Hughes Medical Institute (to MC Wang), and the Quigley Endowed Chair in Cardiology (to XHT Wehrens). This project was supported by the Genetically Engineered Rodent Models Core at Baylor College of Medicine, which is funded in part by the NIH Cancer Center Grant (P30 CA125123). The project was also supported by the Mouse Metabolism and Phenotyping Core at Baylor College Medicine, supported in part from NIH (UM1HG006348, R01DK114356, R01HL130249) and using instrument purchased with NIH funding (S10OD032380).

## Author Contributions

SK Lahiri: conceptualization, data curation, formal analysis, investigation, visualization, methodology, and writing—original draft.
F Jin: lipidomics experiment, formal analysis and investigation.
Y Zhou: formal analysis and investigation.
AP Quick: investigation.
CF Kramm: investigation.
MC Wang: conceptualization, supervision, funding acquisition, and project administration.
XHT Wehrens: conceptualization, supervision, funding acquisition, project administration, and writing—review and editing.

## Conflict of Interest Statement

XHT Wehrens serves as a consultant for Pfizer and Rocket Pharmaceuticals, in addition to being a founding partner and board member at Elex Biotech Inc. The remaining authors have no relevant disclosures to report.

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
