## [Reviewer comments · Life Science Alliance]

Life Science Alliance

Altered Myocardial Lipid Regulation in Junctophilin-2 Associated Familial Cardiomyopathies

Satadru Lahiri, Feng Jin, Yue Zhou, Ann Quick, Carlos Kramm, Meng Wang, and Xander Wehrens

DOI: <https://doi.org/10.26508/lsa.202302330>

Corresponding author(s): Xander Wehrens, Baylor College of Medicine

Review Timeline:

Submission Date:	2023-08-22
Editorial Decision:	2023-09-28
Revision Received:	2024-02-19
Editorial Decision:	2024-02-22
Revision Received:	2024-02-22
Accepted:	2024-02-23

Transaction Report:

September 28, 2023

Re: Life Science Alliance manuscript #LSA-2023-02330-T

Prof Xander H.T. Wehrens
Baylor College of Medicine
Molecular Physiology and Biophysics
1 Baylor Plaza
Houston, TX 77030

Dear Dr. Wehrens,

Thank you for submitting your manuscript entitled "Altered Myocardial Lipid Regulation in Junctophilin-2 Associated Familial Cardiomyopathies" to Life Science Alliance. The manuscript was assessed by expert reviewers, whose comments are appended to this letter. We invite you to submit a revised manuscript addressing the Reviewer comments.

Thank you for this interesting contribution to Life Science Alliance. We are looking forward to receiving your revised manuscript.

Sincerely,

B. MANUSCRIPT ORGANIZATION AND FORMATTING:

Reviewer #1 (Comments to the Authors (Required)):

The manuscript Altered Myocardial Lipid Regulation in Junctophilin-2 Associated Familial Cardiomyopathies by Lahiri et al., is an original research manuscript and focuses on the role of the Junctophilin-2 (JPH2), generally known to connect plasma membrane and ER, in regulating lipids in the left ventricles of the heart. The article utilizes two different clinically relevant mutations to study their effects on the levels of different classes of lipids. The manuscript is well-written, and the data presented is robust. A few minor comments need to be answered before the manuscript can be accepted for publication:

- The authors need to indicate the amino acid in mice Jph2 that corresponds to the human E641* variant. If the position of this amino acid is the same in both species, then the authors should indicate this.
- In Figure 1A, the authors should include a representation of the wild-type form of Jph2 along with its mutant forms to make the figure more relevant. A representation of wild-type human JPH2 could also be included to better understand the sequence variation between the two species.
- Since the authors generated these mice, authors need to include genomic DNA analysis from wild-type, heterozygous mutant mice and homozygous mutant mice confirming these mutations. This data could be added as a supplementary figure.
- For Figure 2C, the authors could utilize a lower exposure image for JPH2. Since the same antibody was used for detecting Jph2 levels in both Figure 2A and 2C, the authors need to crop the blots similarly for both mutants. Also, mice Jph2 are mentioned differently in the text as compared to figures and figure legends, the authors should maintain consistent labeling of the protein throughout the manuscript.
- The authors need to demonstrate that E641* mutation results in complete loss of expression of the protein at a cellular level. Also, they need to justify their statement of attributing the loss of Jph2 to the inviability of the mice using other studies with similar results in mice or other species.
- All figures should be arranged as per their first citation in the text.
- In Figure 3A, the authors could depict the changes they observed in current Figure 1.
- Each sub-figure needs to be cited at least once in the text.

'Referee Cross-Comments'

The comments by other two reviewers are valid and need to be answered by the authors before proceeding with the manuscript.

Reviewer #2 (Comments to the Authors (Required)):

This paper describes an unbiased lipidomic screen of 3 models for cardiomyopathies. Two related models for hypertrophic cardiomyopathy (Junctophilin 2 mutation A399S) and one for dilated cardiomyopathy (JPH2 mutation E641*) . The lipidomic analysis showed differences between the groups which might be physiological relevant. However due to the complexity of lipidomics, lipid metabolism and the way the authors depict their data no clear new insights conclusions can be gained
Main points.

1. The data only show differences in the amounts the various lipids between the groups but no complete data set is presented to give an idea how quantitatively abundant the differing lipid species are. i.e are this traces or major components of the lipidome. So a complete data set should accompany this paper for better evaluation and appreciation by the community
2. As the list of different lipids contain many very obscure lipids (uneven and/or very short chain length) the author should present evidence for a correct annotation of these (minor? See point 1) lipids
3. Calculation of the significance levels of the various should be explained. How was corrected for the number of different lipids analyzed?

Reviewer #3 (Comments to the Authors (Required)):

The study by Lahiri et al. aimed to investigate the effect of JPH2 variants identified in patients with HCM and DCM, respectively, on myocardial lipid profiles. Using unbiased myocardial lipidomic analysis, the authors showed a significantly reduced level of total unsaturated fatty acids, ceramides, and various phospholipids in both mice with HCM and DCM. The authors also showed a significant increase in di- and triglycerides and a decrease in co-enzyme only in mice with HCM. Further in-depth analysis

revealed significantly altered metabolites from all the lipid classes with similar or opposing trends in Jph2 mutant mice with HCM or DCM. The authors concluded that distinct Jph2 variants cause alterations in the cardiac lipid composition.

I have the following major concerns regarding the study and its conclusion:

Critique:

Major concerns:

- 1- My major concern is that these preliminary data do not provide significant insights into the role of JPH2 in modifying cardiac energy metabolism and structure. These data show that JPH2 variants are bad news to the heart function and structure, which are not novel insights and have already been shown before.
- 2- The changes in lipid classes in A399S-het, A399S-homo and E641*-het mice do not clearly explain what these changes mean or why these changes are happening. For example, the decrease in the fatty acyl group in all these phenotypes could mean an acceleration in the processing of these metabolites and an increase in fatty acid oxidation, or it could indicate a decrease in fatty acid uptake or esterification as a feedback mechanism to a decrease in fatty acid oxidation. Therefore, the data in this manuscript do not provide meaningful new insights without direct measurements of fatty acid uptake and oxidation in these phenotypes.
- 3- Fatty acid utilization in the heart is tightly regulated and influenced by glucose utilization. Therefore, direct measurement of glucose uptake and oxidation in these phenotypes will be necessary to understand the effect of JPH2 dysfunction on cardiac energy metabolism.
- 4- How does modifying A399S and E641 impact JPH2 activity?
- 5- The study lacks evidence that JPH2 variants directly influence cardiac lipid metabolism.
- 6- It is unclear why the authors cited (Iopaschuk et al 2021) after this sentence "Next, we assessed cardiac function in the mutant mice to determine whether it mimics the clinical phenotype observed in human patients carrying the corresponding JPH2 variants"?

Response to Reviewers' Comments**Life Science Alliance manuscript #LSA-2023-02330-T****"Altered Myocardial Lipid Regulation in Junctophilin-2 Associated Familial Cardiomyopathies"****EDITORS****-- A letter addressing the reviewers' comments point by point.**

Please see below.

A word text file is provided.

For the first time, an unbiased study identified variations in myocardial lipid compositions in genetic models of hypertrophic and dilated cardiomyopathy associated with the non-sarcomere protein junctophilin-2.

The authors accept the publication fee.

We have added the uncropped gel images as a source data file. We have also added the lipidomics dataset.

Reviewer #1 (Comments to the Authors (Required)):**The manuscript Altered Myocardial Lipid Regulation in Junctophilin-2 Associated Familial Cardiomyopathies by Lahiri et al., is an original research manuscript and focuses on the role of the Junctophilin-2 (JPH2), generally known to connect plasma membrane and ER, in regulating lipids in the left ventricles of the heart. The article utilizes two different clinically relevant mutations to study their effects on the levels of different classes of lipids. The manuscript is well-written, and the data presented is robust. A few minor comments need to be answered before the manuscript can be accepted for publication:****1: The authors need to indicate the amino acid in mice Jph2 that corresponds to the human E641* variant. If the position of this amino acid is the same in both species, then the authors should indicate this.**

Response: We thank the reviewer for carefully evaluating of our manuscript. The amino acid E641 is conserved in both human and mouse JPH2. Please refer to the JPH2 protein sequence alignment (aa594-650) sourced from NCBI BlastP (below); the E641 residue is highlighted within a red rectangle. In the revised manuscript, we have added a sentence “Interestingly, the position of the E641 residue is same in both mouse and human JPH2” in the Introduction section.

Human JPH2	594	SAPSSPATAPL---QAPTLRGPEPARETPAKLEPKPIIPKAEPRAKARKTEARGLTKAGA	650
Mouse JPH2	593	SAPSPVSATVPEEEFPAPRSPVPAKQ--ATLEPKPIVPKAEPAKARKTEARGLSKAGA	650

2: In Figure 1A, the authors should include a representation of the wild-type form of Jph2 along with its mutant forms to make the figure more relevant. A representation of wild-type human JPH2 could also be included to better understand the sequence variation between the two species.

Response: We thank the reviewer for this comment. Figure 1A has been revised by include the WT sequence.

3: Since the authors generated these mice, authors need to include genomic DNA analysis from wild-type, heterozygous mutant mice and homozygous mutant mice confirming these mutations. This data could be added as a supplementary figure.

Response: We appreciate the reviewer's feedback. To address the comment, we have extracted, amplified, and sequenced the genomic region encompassing the A399S mutation in JPH2 from wild type (WT), heterozygous (A399S het), and homozygous (A399S homo) mice. The resulting data, demonstrating the A to S mutation at position 399 in JPH2, is now presented in a new Supplemental Figure 1 that includes both WT and A399S homozygous mice.

Additionally, we have sequenced the E641 region from WT and E641* heterozygous mice. Due to the presence of mixed alleles, the 'T' base insertion was not distinctly visible in the heterozygous sequence. To overcome this, we employed TA cloning to isolate the WT and E641* alleles from the E641* heterozygous mice. Subsequent sequencing of the isolated mutant clone confirmed the insertion of the 'T' base, which introduces an early stop codon (*) in the E641 mice. This sequencing data has also been incorporated into Supplemental Figure 1 alongside the A399S mutation sequencing.

4: For Figure 2C, the authors could utilize a lower exposure image for JPH2. Since the same antibody was used for detecting Jph2 levels in both Figure 2A and 2C, the authors need to crop the blots similarly for both mutants. Also, mice Jph2 are mentioned differently in the text as compared to figures and figure legends, the authors should maintain consistent labeling of the protein throughout the manuscript.

Response: We appreciate the feedback provided by the reviewer. In response, we have updated Figure 2 (New Fig 1) by substituting the earlier E641* JPH2 blot with a version that has lower exposure. Additionally, we have trimmed the E641* blot to match the presentation of the A399S JPH2 blot. The original full-sized blots are also supplied as supplementary material. In accordance with the reviewer's suggestions, we have also corrected the notation from 'Jph2' to 'JPH2' in the text to keep it consistent.

5: The authors need to demonstrate that E641* mutation results in complete loss of expression of the protein at a cellular level. Also, they need to justify their statement of attributing the loss of Jph2 to the inviability of the mice using other studies with similar results in mice or other species.

Response: Thank you for the reviewer comment. Our findings show a 50% reduction of JPH2 protein in E641* heterozygous mice. Additionally, our JPH2-E641* mouse colony has not yielded any homozygous mice. A prior

study by our team showed that cardiac-specific downregulation of JPH2 by 70-80% using shJPH2 at embryonic day 10.5 resulted in cardiac dilation and premature death within 15 days after birth (PMID: 23715556). Combining these observations leads us to hypothesize that JPH2 levels could be nearly completely suppressed in homozygous mice, potentially causing embryonic fatality, and thus explaining their absence in our colony. We concur with the reviewer, and we believe that a comprehensive embryonic investigation within the E641* colony is essential to unravel the impact of this specific mutation on the JPH2 protein.

6: All figures should be arranged as per their first citation in the text.

Response: We appreciate your feedback. Figures 1 and 2 have been revised to align with the narrative in the results section. Corresponding modifications have been made to the figure captions and the manuscript text. Additionally, we have rearranged several sentences in the results section to reference Figure 3E prior to Figure 4. Upon review of the article, we have verified and ensured that the figures are presented in the order of their mention in the manuscript.

7: In Figure 3A, the authors could depict the changes they observed in current Figure 1.

Response: We thank the reviewer for the comments. In response, we have enlarged the left ventricular dimension in the E641* heart illustration to accurately represent the dilated cardiomyopathy phenotype characteristic of this model.

8: Each sub-figure needs to be cited at least once in the text.

Response: We thank the reviewer for this comment. We have reviewed our manuscript and have now included Figure 5A-C and the supplementary subfigures that were previously omitted.

Reviewer #2 (Comments to the Authors (Required)):

This paper describes an unbiased lipidomic screen of 3 models for cardiomyopathies. Two related models for hypertrophic cardiomyopathy (Junctophilin 2 mutation A399S) and one for dilated cardiomyopathy (JPH2 mutation E641*). The lipidomic analysis showed differences between the groups which might be physiological relevant. However due to the complexity of lipidomics, lipid metabolism and the way the authors depict their data no clear new insights conclusions can be gained.

Main points.

1: The data only show differences in the amounts the various lipids between the groups but no complete data set is presented to give an idea how quantitatively abundant the differing lipid species are. i.e are this traces or major components of the lipidome. So a complete data set should accompany this paper for better evaluation and appreciation by the community

2: As the list of different lipids contain many very obscure lipids (uneven and/or very short chain length) the author should present evidence for a correct annotation of these (minor? See point 1) lipids

Response: We agree with these comments. Consequently, we have now added the overall lipidomics data, which illustrates the prevalence of various lipids and their respective fold changes across different groups. The included list details the lipids, specifying ion heads. It should be noted that for certain lipids, our identification is limited to the head group and fatty acyl chains. In cases where double bonds are present, we may lack sufficient fragment information to determine the specific location of these double bonds.

3: Calculation of the significance levels of the various should be explained. How was corrected for the number of different lipids analyzed?

Response: We thank the reviewer for the comment. We have quantified the statistical differences in various lipid groups and lipid metabolites across different mouse cohorts, employing ANOVA (For A399S 3 groups) or T-tests (E641* 2 groups) in accordance with recommendations from the Lipidsearch software. However, we have not implemented a correction for the multiplicity of lipid analyses, as we did not assess the global alterations in the lipidomic profiles between our mouse models.

Reviewer #3 (Comments to the Authors (Required)):

The study by Lahiri et al. aimed to investigate the effect of JPH2 variants identified in patients with HCM and DCM, respectively, on myocardial lipid profiles. Using unbiased myocardial lipidomic analysis, the authors showed a significantly reduced level of total unsaturated fatty acids, ceramides, and various phospholipids in both mice with HCM and DCM. The authors also showed a significant increase in di- and triglycerides and a decrease in co-enzyme only in mice with HCM. Further in-depth analysis revealed significantly altered metabolites from all the lipid classes with similar or opposing trends in Jph2 mutant mice with HCM or DCM. The authors concluded that distinct Jph2 variants cause alterations in the cardiac lipid composition.

I have the following major concerns regarding the study and its conclusion:

Critique:

Major concerns:

1: My major concern is that these preliminary data do not provide significant insights into the role of JPH2 in modifying cardiac energy metabolism and structure. These data show that JPH2 variants are bad news to the heart function and structure, which are not novel insights and have already been shown before.

Response: We thank the reviewer for the comment. We acknowledge that our study does not establish a direct causal relationship between JPH2 and cardiac metabolism. Nonetheless, this research is among the pioneering efforts to conduct lipidomic profiling directly from cardiac tissues, moving beyond the convention of plasma profiling. It also marks the first comparative lipidomic analysis between two distinct cardiac disease models, both originating from different mutations in the same gene, JPH2. In addition, we have developed two novel JPH2 mutant knock-in mouse models representing hypertrophic and dilated cardiomyopathy, which will be valuable for future research. Our lipidomic evaluation has uncovered both shared and unique alterations in lipid metabolites between these two cardiomyopathy models. Notably, a dose-dependent effect was observed in JPH2 A399S heterozygous and homozygous mice, hinting at JPH2's potential role. While utilizing this study as a foundation, comprehensive lipid flux analyses in these models may uncover more complex associations between JPH2 and cardiac metabolism in hereditary cardiomyopathies. We recognize that such a flux study would be compelling; however, it falls outside the scope of the current paper.

2: The changes in lipid classes in A399S-het, A399S-homo and E641*-het mice do not clearly explain what these changes mean or why these changes are happening. For example, the decrease in the fatty acyl group in all these phenotypes could mean an acceleration in the processing of these metabolites and an increase in fatty acid oxidation, or it could indicate a decrease in fatty acid uptake or esterification as a feedback mechanism to a decrease in fatty acid oxidation. Therefore, the data in this manuscript do not provide meaningful new insights without direct measurements of fatty acid uptake and oxidation in these phenotypes.

Response: We thank the reviewer for the comment. We concur that the observed changes in lipidomic profile in our mouse models warrants a thorough investigation into cardiac metabolism. In response, we conducted a Seahorse assay on adult cardiomyocytes derived from our models to evaluate their oxygen consumption rate (OCR).

As shown in the figure provided for the reviewer's reference, we began with a mito stress test (**A**). Despite noting a decrease in metabolic activity in both A399S and E641* mutant cardiomyocytes compared to the wild type (WT), the differences did not reach statistical significance, likely due to variability in the data. We were limited to using cells from a single mouse from each group on a 96-well plate, thus we completed three separate Seahorse assays to include cells from a minimum of three mice per group. Significant variability in OCR data was noted across these experiments, even though around 6000 cardiomyocytes were consistently seeded per well. We suspect that the isolated cardiomyocytes began deteriorating within 3-4 hours post-isolation, which is a critical timeframe for this experiment, indicating a need for further methodological refinement as adult cardiomyocytes are not amenable to prolonged culture, which would be preferable for optimized Seahorse assay outcomes.

Additionally, we introduced either BSA or BSA-conjugated palmitate to the cardiomyocytes prior to the Seahorse assay to gauge fatty acid absorption. Our findings indicated a marked elevation in OCR in WT cells following palmitate addition. However, this increase in OCR was not mirrored significantly in any of the mutant cardiomyocytes, implying a possible limitation in fatty acid absorption in these cells. In summary, our data suggest a pattern of diminished cardiac metabolism and fatty acid intake in the JPH2 mutant cardiomyocytes, which might account for the decreased fatty acyl groups observed in our lipidomic profile. Nonetheless, further refinement of our assay and additional experiments are necessary to draw definitive conclusions.

[Figure removed by editorial staff per authors' request]

3: Fatty acid utilization in the heart is tightly regulated and influenced by glucose utilization. Therefore, direct measurement of glucose uptake and oxidation in these phenotypes will be necessary to understand the effect of JPH2 dysfunction on cardiac energy metabolism.

Response: We thank the reviewer for this comment. We have applied Etomoxir, a fatty acid oxidation inhibitor, and UK5099, a glucose oxidation inhibitor, to the isolated cardiomyocytes derived from our disease models to investigate both fatty acid and glucose oxidation pathways. Nonetheless, in the Seahorse assays, the impact of these inhibitors on the cardiomyocytes was not distinctly evident. As previously acknowledged, further optimization of these assays is required for conclusive results, which we consider to be outside the purview of this current manuscript.

4: How does modifying A399S and E641 impact JPH2 activity?

Response: We thank the reviewer for this comment. The A399 residue is situated within the alpha-helix region of JPH2, while the E641 residue is found just before the transmembrane domain at the C-terminus of the protein. The specific functions of these residues are not fully understood at present. Previous research has indicated that the alanine-rich segment spanning amino acids 331-405 in mouse JPH2 is essential for the protein's ability to bind DNA (PMID: 30409805, 35665900). Another study has highlighted the critical nature of the human JPH2 A405 site (equivalent to A399 in mouse JPH2) for the formation of nuclear liquid droplets by the human protein (PMID:34062390). In our research, the A399S mutants showed no alteration in JPH2 protein levels, implying that the A399 site is pivotal for JPH2's function. Further investigation is necessary to elucidate the impact of this site on the structure and function of JPH2. On the other hand, the E641* mutation introduces a premature STOP codon at the position of the E641 residue, theoretically resulting in a truncated JPH2 protein that lacks the transmembrane domain. However, our western blot analysis of heart tissue from E641* mice did not detect any truncated JPH2 protein, which suggests that this mutation may cause the protein to degrade. Furthermore, we observed a roughly 50% decrease in full-length JPH2 protein levels in heterozygous E641* mice, supporting the notion that the truncated protein is unstable, leading to a reduced amount of overall JPH2 in these mice. This instability could explain why homozygous E641* mice are not viable, likely due to the critical loss of full-length JPH2 protein. Based on these findings, we propose that these two mutations exert distinct influences on JPH2, as evidenced by varying cardiac effects and differences in lipidomic profiles.

5: The study lacks evidence that JPH2 variants directly influence cardiac lipid metabolism.

Response: We thank the reviewer for this comment. We concur that the findings presented in this manuscript do not assert a direct causative link between specific JPH2 variants and lipid metabolism alterations. Nonetheless, our data does indicate that JPH2 dysfunction precipitates significant changes in the lipidomic profiles. The observed dose-dependent effects on lipid regulation in both heterozygous and homozygous JPH2 A399S models imply a role for JPH2. Notably, distinct lipidomic signatures correlated with different forms of cardiomyopathy were identified in each of the JPH2 mutant models. To further discern how alterations in JPH2 directly affect cardiac lipid metabolism, future studies would require the reestablishment of normal JPH2 expression in the E641* model or restoration of JPH2 functionality in the A399S model, with the aim of reversing the lipidomic changes observed. However, we believe that is beyond the scope for this manuscript.

6: It is unclear why the authors cited (Iopaschuk et al 2021) after this sentence "Next, we assessed cardiac function in the mutant mice to determine whether it mimics the clinical phenotype observed in human patients carrying the corresponding JPH2 variants"?

Response: We are thankful for the reviewer's observation. Upon review, we identified an Endnote error and have made the necessary correction. Additionally, we have carefully examined the manuscript for any other misplaced references that may have occurred during the revision process and have rectified these as well.

February 22, 2024

RE: Life Science Alliance Manuscript #LSA-2023-02330-TR

Prof. Xander H.T. Wehrens
Baylor College of Medicine
Molecular Physiology and Biophysics
1 Baylor Plaza
Houston, TX 77030

Dear Dr. Wehrens,

Thank you for submitting your revised manuscript entitled "Altered Myocardial Lipid Regulation in Junctophilin-2 Associated Familial Cardiomyopathies". We would be happy to publish your paper in Life Science Alliance pending final revisions necessary to meet our formatting guidelines.

- please be sure that the authorship listing and order is correct
- please upload all figure files as individual ones, including the supplementary figure files; all figure legends should only appear in the main manuscript file
- figure S6 has only one panel, so there is no need to label it as A. -- please correct the figure, its legend, and callout
- please add the Twitter handle of your host institute/organization as well as your own or/and one of the authors in our system
- please add an Author Contributions section to your main manuscript text
- please add a conflict of interest statement to your main manuscript text
- please use the [10 author names et al.] format in your references (i.e., limit the author names to the first 10)
- please add your main, supplementary figure, and table legends to the main manuscript text after the references section
- remove figures from the manuscript text...they should be provided separately, and their legend should appear only in the manuscript text after the references section
- please add callouts for Figure S3A-B to your main manuscript text

A. FINAL FILES:

B. MANUSCRIPT ORGANIZATION AND FORMATTING:

Sincerely,

February 23, 2024

RE: Life Science Alliance Manuscript #LSA-2023-02330-TRR

Prof. Xander H.T. Wehrens
Baylor College of Medicine
Molecular Physiology and Biophysics
1 Baylor Plaza
Houston, TX 77030

Dear Dr. Wehrens,

Thank you for submitting your Research Article entitled "Altered Myocardial Lipid Regulation in Junctophilin-2 Associated Familial Cardiomyopathies". It is a pleasure to let you know that your manuscript is now accepted for publication in Life Science Alliance. Congratulations on this interesting work.

DISTRIBUTION OF MATERIALS:

Again, congratulations on a very nice paper. I hope you found the review process to be constructive and are pleased with how the manuscript was handled editorially. We look forward to future exciting submissions from your lab.

Sincerely,
